# Cutaneous information processing differs with load type during isometric finger abduction

Keisuke Yunoki[1], Tatsunori Watanabe[1,2], Takuya Matsumoto[1,3], Takayuki Kuwabara[1,4], Takayuki Horinouchi ●[1], Kanami Ito[1], Haruki Ishida[1], Hikari Kirimoto ●[1] *

1 Department of Sensorimotor Neuroscience, Graduate School of Biomedical and Health Sciences, Hiroshima University, Hiroshima, Japan, 2 Faculty of Health Sciences, Aomori University of Health and Welfare, Aomori, Japan, 3 Research Fellow of Japan Society for the Promotion of Science, Chiyoda-ku, Japan, 4 Department of Rehabilitation, Uonuma Kikan Hospital, Minamiuonuma, Niigata, Japan

* hkirimoto@hiroshima-u.ac.jp

**Data Availability Statement:** All relevant data are within the paper and its Supporting Information files.

## Abstract

During submaximal isometric contraction, there are two different load types: maintenance of a constant limb angle while supporting an inertial load (position task) and maintenance of a constant force by pushing against a rigid restraint (force task). Previous studies demonstrated that performing the position task requires more proprioceptive information. The purpose of this study was to investigate whether there would be a difference in cutaneous information processing between the position and force tasks by assessing the gating effect, which is reduction of amplitude of somatosensory evoked potentials (SEPs), and cutaneomuscular reflex (CMR). Eighteen healthy adults participated in this study. They contracted their right first dorsal interosseous muscle by abducting their index finger to produce a constant force against a rigid restraint that was 20% maximum voluntary contraction (force task), or to maintain a target position corresponding to 10˚ abduction of the metacarpophalangeal joint while supporting a load equivalent to 20% maximum voluntary contraction (position task). During each task, electrical stimulation was applied to the digital nerves of the right index finger, and SEPs and CMR were recorded from C3' of the International 10–20 system and the right first dorsal interosseous muscle, respectively. Reduction of the amplitude of N33 component of SEPs was significantly larger during the force than position task. In addition, the E2 amplitude of CMR was significantly greater for the force than position task. These findings suggest that cutaneous information processing differs with load type during static muscle contraction.

## Introduction

Sensorimotor modulation is a process by which the motor system continuously elaborates sensory afferents in order to enhance the execution of fine motor activities [1]. In particular, somatosensory information, especially from group Ia and Aβ fibers, is essential for performing daily tasks and dexterous manual movements. Group Ia fibers originate from muscle spindles

**Funding:** This work was partially supported by a grant from the Japan Society for the Promotion of Science (https://www.jsps.go.jp/) to TW (No. 22K17777), TM (No. 20J21369), and HK (No. 22H03454). The funders had no role in study design, data collection and analysis, decision to publish, or preparation of the manuscript.

**Competing interests:** The authors have declared that no competing interests exist.

and carry information about the position and velocity of a moving joint [2] while Aβ fibers originate from cutaneous mechanoreceptors and carry information about skin strain, stretch, and vibration [3,4]. As a practical example to show its importance, patients with sensory deficits have impairments in maintaining constant motor output [5–7], resulting in a difficulty in performing daily activities, such as writing with a pen and holding a cup. Previous studies have demonstrated that improvements in sensory performance by somatosensory discrimination training (discrimination of touch or limb position) after stroke can be observed with related untrained stimuli from the same modality but not with ones from the other modalities [8,9]. Therefore, interventions for somatosensory impairments resulting from central nervous disorders should target the modality which patients have impaired with. For this reason, it is important to determine in what situation cutaneous information is required more than proprioceptive information, and vice versa. However, in daily life, both Aβ and Ia fibers are activated by movements [10] and these two kinds of information are important for precise movement execution. Therefore, it is difficult to set up an assignment that isolates their respective contributions.

In experimental environments, modality-specific somatosensory information processing, particularly proprioceptive information processing, has been assessed in isometric contraction tasks with different load types, namely force and position tasks [11]. Force task requires subjects to maintain a constant force by pushing against a rigid restraint. In contrast, position task requires subjects to keep a constant limb angle while supporting an inertial load. Although these two tasks require a similar net muscle torque, their underlying neural control mechanisms have been shown to be different. For example, amplitudes of short latency reflex (SLR) and long latency reflex (LLR) were larger in the position task as compared with the force task [11–14]. Also, heteronymous monosynaptic Ia facilitation was greater while homonymous inhibition was smaller in the position than force task [15–17]. Furthermore, the rate of motor unit recruitments was greater during the position than force task [18,19]. These findings suggest that maintaining a constant limb angle while supporting an inertial load may require proprioceptive information more than cutaneous information [20]. Determining which task requires more cutaneous information may contribute to providing basic data required for developing effective exercise prescriptions for rehabilitation of patients with central nervous disorders who suffer impaired superficial sensory perception. However, it is currently unclear how cutaneous information processing differs between the position and force tasks.

Somatosensory evoked potentials (SEPs) or somatosensory evoked magnetic fields (SEFs), and cutaneomuscular reflex (CMR) induced by digital nerve stimulation with ring-type electrodes have been used to assess the processing of cutaneous information in the central nervous system. Some previous studies have reported that amplitudes of short-latency components of SEPs and SEFs were reduced during movements [21,22], which is known as "gating" [23–27]. The functional role of this SEP gating is considered to filter out irrelevant or redundant somatosensory information, ensuring the processing of the relevant feedback [23,26,28]. Kirimoto et al. [14] demonstrated that reduction of amplitude of P45 component of SEPs was larger during the position than force task with the index finger abductor muscles when the ulnar nerve (innervating the agonist muscle in the task), but not the median nerve, was stimulated. However, a difference in SEP gating with digital nerve stimulation between the position and force tasks has yet to be explored.

Electrical stimulation of the digital nerves during sustained voluntary contraction can induce reflex modulation of ongoing electromyographic (EMG) activity in the contracting muscle, which is known as the CMR [29–32]. The CMR recorded from the first dorsal interosseous (FDI) muscle typically consists of an early period of facilitation (E1, 30–40 ms) followed by a period of inhibition (I1, 40–50 ms) and more prominent second facilitation (E2,

50–80 ms) [31,33]. The E1 component of CMR is believed to be mediated via oligosynaptic spinal pathway [30,34], whereas the I1 and E2 components are considered to involve a transcortical pathway [30,32,35–38]. A unique feature of the E2 component is that it is strongly modulated during movements in a task-dependent manner [21,31,39,40]. The functional role of the CMR is thought to help execution of higher motor performance by reactively controlling the muscle activity based on cutaneous feedback [40–42]. When static forces are applied to an object by fingertips, a number of SA-I (slow-adapting type I) and SA-II afferents discharge continuously, indicating that cutaneous information about the contact is provided to the central nervous system [3]. Given this finding, it is reasonable to assume that, since changes in the length of muscle and joint angle are minimal during the force task, maintaining a constant force requires cutaneous more than proprioceptive information.

Accordingly, the purpose of this study was to determine whether cutaneous information processing differs with load type. To this end, we recorded SEPs elicited by digital nerve stimulation during the position and force control tasks with the index finger abduction, and compared the magnitude of SEP gating. In addition, we compared the amplitude of CMR recorded from the FDI muscle between these two tasks. We hypothesized that the magnitude of SEP gating and the amplitude of E2 component of CMR would be greater during the force than position task.

## Methods

### Participants

Eighteen healthy adults (15 males and 3 females, 21–35 years old) participated in this study. All participants were strongly right-handed according to the Oldfield inventory scores (score range: 0.9–1.0) [43], and had normal or corrected-to-normal vision. Written informed consent was obtained from all participants before beginning the experiment. This study was approved by the Ethics Committee of Hiroshima University (No. E-2261) and was conducted following the Declaration of Helsinki.

### Experimental setup

Participants were seated in a chair with the right hand positioned in the custom-designed apparatus, which was used in our previous study (Fig 1) [14]. The custom-designed device consisted of a wheel connected to a force transducer (TU-QR, TEAC, Tokyo, Japan) or inertial load by means of pulley and nylon line. The right shoulder was slightly abducted (10–20˚), the right elbow joint was flexed at 110˚, and the right forearm and wrist were held in the neutral position. The index finger was attached to a bar that was connected to the rotating wheel so that the rotational axis of the metacarpophalangeal joint approximated that of the wheel [44]. Furthermore, the extension-flexion movements of the metacarpophalangeal and interphalangeal joints of the index finger were restricted; thus, the participants were allowed to perform only abduction-adduction of the metacarpophalangeal joint. The thumb was restrained at 45˚ of abduction, and the middle, ring and little fingers were fixed to the device with the metacarpophalangeal and interphalangeal joints fully extended. All participants performed two submaximal contraction tasks at a similar torque level. In the force task (Fig 1A), 20% maximum voluntary contraction (MVC) force level was maintained with the metacarpophalangeal joint at 10˚ of abduction. Meanwhile, in the position task (Fig 1B), the metacarpophalangeal joint angle was not fixed, and participants were required to maintain a target position corresponding to 10˚ abduction of the metacarpophalangeal joint while supporting a load equivalent to 20% MVC. The abduction angle of the metacarpophalangeal joint during the position task was measured with an electro-goniometer (SG65, Biometrics, Gwent, UK) taped to the dorsal

(A) Force task (B) Position task

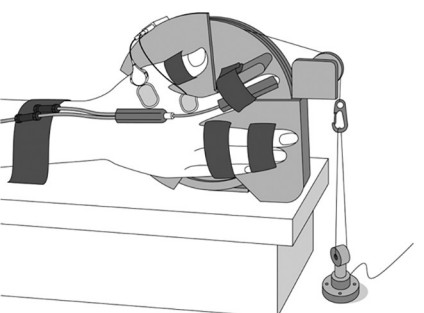 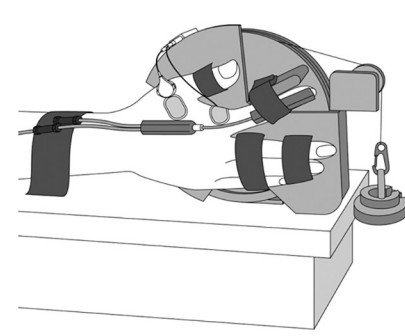

**Fig 1.** Illustration of the experimental setup for the force task (A) and the position task (B). The right hand was placed on the custom-designed apparatus which consisted of a wheel connected to a force transducer (A) or inertial load (B) by means of pulley and nylon line. The abduction angle of the metacarpophalangeal joint during the position task was measured with an electro-goniometer taped to the dorsal surface of the right hand.

surface of the right hand. Visual feedbacks of the joint angle and the abduction force were displayed on a monitor during the position and force tasks, respectively. The gain of the visual feedback was equal to 2.5%/cm of the maximal performance range, operationally defined as a MVC for the force task and full range of motion about the metacarpophalangeal joint for the position task [14,45].

Signals from the force transducer and the electro-goniometer were low-pass filtered at 50 Hz, digitized at 10 kHz (PowerLab, AD Instruments, New South Wales, Australia), and stored on a personal computer for off-line analysis (LabChart 8, AD Instruments, New South Wales, Australia).

## Protocol

At the beginning of the session, we assessed MVC of the right index finger abduction. Participants were instructed to gradually increase force from 0 to maximum over 3 s, and then to hold the maximum force for 3 s with verbal encouragement. MVC was recorded at least three times with resting periods of 90 s between trials. If MVC forces were within 5% of each other, we adopted the highest value as the maximum. If not, additional trials were performed until the 5% criterion was achieved. The maximum value was used as a reference for the submaximal contractions and for the EMG normalization. Then, subjects executed the position and force tasks for approximately 90 s (divided into three blocks of 30 s). These data were used to confirm the reliability of the EMG activity of the right FDI muscle between both tasks.

After the assessment of the reliability of the EMG activity, SEP from C3' (2 cm posterior to C3 of the International 10–20 system) and CMR from the right FDI muscle were recorded. Both SEP and CMR were elicited by digital nerve stimulation to the right index finger. The SEP was recorded during three conditions: resting (as control) and the position and force tasks. The CMR was recorded during two conditions: the position and force tasks. The measurements of SEP and CMR were conducted in separate submaximal contraction trials. Each trial lasted approximately 50–60 s and was separated by 60 s of rest to avoid fatigue (Fig 2). Participants performed each of the three conditions three times for the SEP measurement, and each of the two conditions once for the CMR measurement. The order of the conditions was randomized across participants. To minimize the influence of transient force fluctuations, stimuli were delivered when the force and position signals reached their respective targets and

(A) SEPs recordings

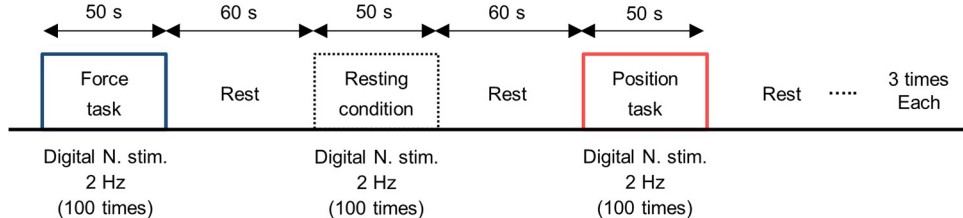

(B) CMR recordings

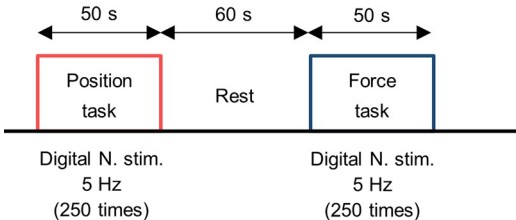

**Fig 2. Experimental procedure.** Experimental procedure for SEP (A) and CMR recordings (B). Digital nerve stimuli were delivered a total of 300 and 250 times during SEP and CMR recordings, respectively.

were maintained at a steady state for at least 1 s [14]. During the recordings of SEPs and CMR, we monitored the EMG activity of the right FDI muscle to ensure that it remained equal between the two tasks.

## Recordings of SEPs

The digital nerve of the right index finger was electrically stimulated with ring-type electrodes. The anode and cathode were attached to the intermediate phalanx and proximal phalanx, respectively. The ground electrode was attached to the right wrist using a disposable gel electrode. Stimulus intensity was adjusted to approximately 3 times each participant's sensory threshold. The sensory threshold was defined as the lowest intensity that consistently produces the subtle tactile sensation on the index finger skin. Square-wave pulses with a 0.2 ms pulse width were delivered at a rate of 2 Hz (A total of 300 responses were averaged for each condition) using a constant-current electrical stimulator (Digitimer DS7, Digitimer, Welwyn Garden City, UK). The stimulator was controlled by the LabChart stimulator system (LabChart 8, AD Instruments).

A silver-silver chloride electrode was placed 2 cm posterior to C3 (C3'), based on the International 10–20 system. A reference electrode was placed on the right earlobe. The evoked potentials were amplified (FA-DL-160, 4 Assist, Tokyo, Japan) and bandpass filtered at 1.6–500 Hz, digitized at 10 kHz (PowerLab, AD Instruments), and stored on a personal computer for offline analysis. SEP waveforms were evaluated for 100 ms with a pre-stimulus period of 20 ms.

## Recordings of CMR

CMR responses to the digital nerve stimulation were recorded from the right FDI muscle. The positions of stimulating and ground electrodes were same as the SEP recording. Stimulus intensity was adjusted to approximately 3 times each participant's sensory threshold. Square-

wave pulses with a 0.2 ms pulse width were delivered at a rate of 5 Hz (A total of 250 responses were averaged for each condition). The electrical stimulation was performed by the same system as in the SEP recording. Surface EMG signals from the right FDI muscle were recorded with disposable silver-silver chloride surface electrodes (1.0 cm diameter). The electrodes were placed over the muscle-belly and distal tendon. EMG signals were amplified (FA-DL-140, 4 Assist, Tokyo, Japan), bandpass filtered at 5–500 Hz, digitized at 10 kHz (PowerLab, AD Instruments), and stored on a personal computer for offline analysis. CMR waveforms were evaluated for 200 ms with a pre-stimulus period of 50 ms.

### Data and statistical analyses

To quantify the maximum EMG activity of the right FDI muscle, EMG activity during MVC trials was rectified and then averaged over a 0.5 s interval centered about the peak EMG. For SEPs, we used the 20 ms period preceding stimulation as the baseline. The peak-to-peak amplitudes of four cortical SEP components (N20, P25, N33 and P45) were analyzed. The amplitude of each component was calculated from the preceding peaks (e.g. the amplitude of P25 was calculated N20-P25 peak-to-peak amplitude). EMG activity during the CMR recording was rectified and averaged for each condition. The magnitude of each of the reflex components (E1, I1 and E2) was calculated by subtracting the mean background EMG (bEMG) activity from the peak amplitude (baseline to peak), and thus was expressed as a percentage of the bEMG activity (%EMG) [46]. The mean bEMG activity was calculated by averaging EMG activity over a 50 ms pre-stimulus period. The bEMG was normalized according to the EMG amplitude at MVC (%MVC).

All data were expressed as the mean ± SEM. SPSS Statistics software version 21 (SPSS; IBM Corp., NY, United States) was used for statistical analysis. Normal distribution of the data was assessed with Shapiro-Wilk test, and if not normal, non-parametric statistical tests were used. The magnitude of bEMG, and amplitudes of E1, I1, and E2 components of the CMR were compared between the position and force tasks using a paired $t$-test (parametric) or Wilcoxon signed-rank test (non-parametric). The amplitudes of the four SEP components were compared among conditions (rest, and position and force tasks) using a repeated-measures analysis of variance (ANOVA). The sphericity of the data was tested with Mauchly's test when conducting ANOVA. Degree of freedom was corrected using Greenhouse-Geisser estimates if sphericity was lacking. *Post-hoc* analysis was performed with Bonferroni's correction for multiple comparisons. Significant level was set at $p < 0.05$.

### Results

Fig 3A shows the grand averaged waveforms of SEPs during the resting, and the position and force tasks. Fig 3B–3E show the mean amplitude of each of the SEP components (N20, P25, N33 and P45). One-way ANOVA revealed a significant main effect of condition for the N20 and N33 components (N20: $F_{(2,34)} = 3.372$, $p = 0.046$, $\eta^2 = 0.166$; N33: $F_{(2,34)} = 4.506$, $p = 0.033$, $\eta^2 = 0.210$), but for the P25 and P45 components (P25: $F_{(2,34)} = 0.840$, $p = 0.441$, $\eta^2 = 0.047$; P45: $F_{(2,34)} = 2.296$, $p = 0.116$, $\eta^2 = 0.119$). *Post-hoc* analysis showed that the amplitude of N20 component did not differ significantly between conditions (rest vs. position, $p = 0.288$; rest vs. force, $p = 1.000$; position vs. force, $p = 0.073$). On the other hand, *post-hoc* analysis showed that the N33 component was significantly smaller during the force than position task (1.12 ± 0.12 μV and 1.40 ± 0.10 μV, respectively; $p = 0.008$). There was no significant difference between the rest and position task ($p = 1.000$), or the rest and force task ($p = 0.076$).

Fig 4A shows the representative waveforms of CMR responses during the position and force tasks. A paired $t$-test revealed that the bEMG activity did not differ between the position

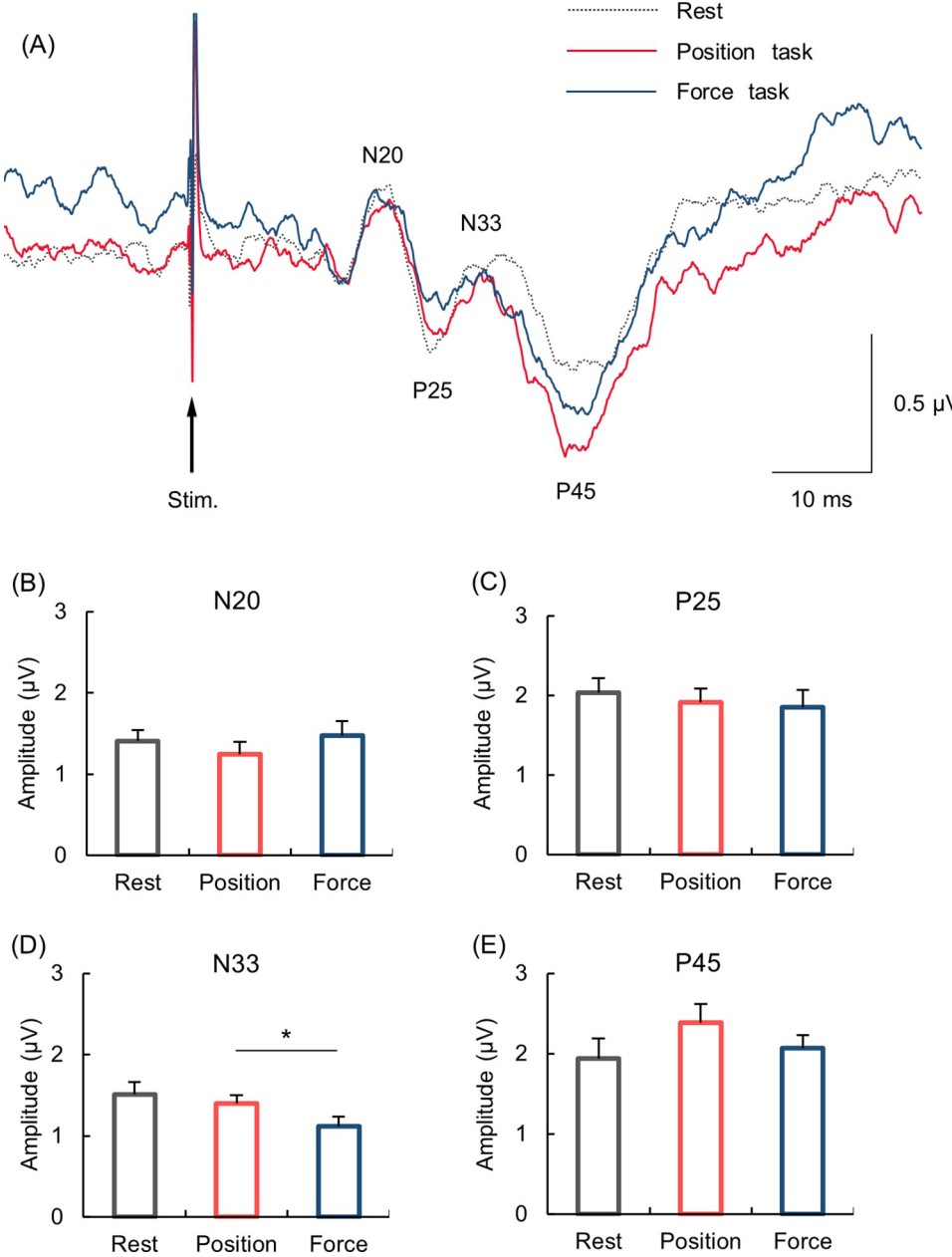

**Fig 3. SEP waveforms and amplitudes of SEP components.** (A) Grand averaged SEP waveforms recorded from C3'
during the resting condition (dotted line), and the position (red line) and force tasks (blue line). (B-E) Mean amplitude
of each SEP component. Error bars indicate the standard error of the mean. The asterisk indicates significant
difference ($p < 0.05$).

and force tasks (17.6 ± 1.1%MVC and 18.0 ± 0.8%MVC, respectively; $p$ = 0.618) (Fig 4B). Fig
4C–4E show the mean amplitude of each of the CMR components (E1, I1 and E2). The ampli-
tudes of E1 and I1 components did not differ between the position and force tasks (E1:
$p$ = 0.306, Wilcoxon signed-rank test; I1: $p$ = 0.170, paired $t$-test). In contrast, the amplitude of
E2 component was significantly greater during the force than position task (36.4 ± 3.3%EMG
and 24.5 ± 2.3%EMG, respectively; $p < 0.001$, Wilcoxon signed-rank test).

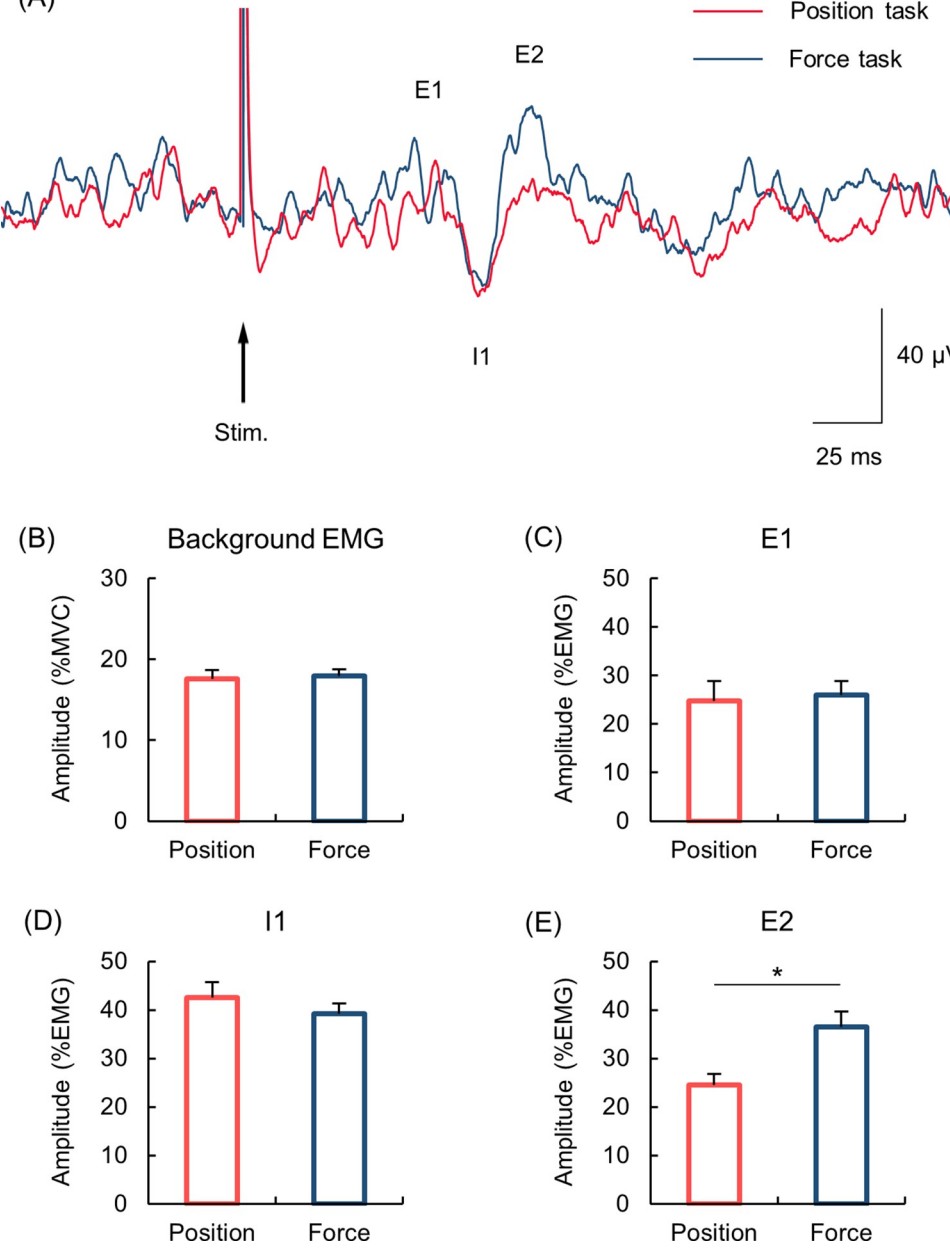

**Fig 4. CMR waveforms and amplitudes of background EMG and CMR components.** (A) Representative CMR waveforms recorded from the right FDI muscle during the position (red line) and force tasks (blue line). (B) Mean amplitude of the background EMG activity. (C-E) Mean amplitude of each CMR component. Error bars indicate the standard error of the mean. The asterisk indicates significant difference ($p < 0.05$).

## Discussion

In this study, we compared cutaneous information processing between the position and force tasks. As a results, reduction of the SEP amplitude (N33) was significantly larger during the force than position task. In addition, the E2 amplitude of CMR was significantly greater for the force than position task. These findings suggest that cutaneous information processing in the central nervous system differs with load type during static muscle contraction.

The gating of SEP amplitude during movement could occur in two possible ways. The first possibility is that the suppression is brought about by interaction between the given sensory signals and the afferent proprioceptive feedback from the muscles, joints, and skin induced by movement, called centripetal gating [26,47,48]. The second possibility is that the suppression occurs as a result of interaction between the given sensory signals and the efferent signals from the motor-related cortical regions, such as the primary and supplementary motor area, called centrifugal gating [27,49]. As combined action of these mechanisms may result in SEP gating in anywhere along the ascending somatosensory pathway and cerebral cortex, it is important to consider both centripetal and centrifugal mechanisms to address our findings. In the present study, gating of the N33 amplitude was greater during the force than position task. This agrees with previous findings showing that the magnitude of SEP gating changed according to the nature and characteristics of the motor task [50–52]. It has been demonstrated that the amplitude of short-latency SEP component is modulated by the centrifugal mechanism depending on the kinesthetic requirements to control movement execution [26,53,54]. Borich et al. [50] proposed that sensitivity to sensory information unrelated to a performing task will be decreased with an increase in demands of task-related sensory processing in order to successfully execute the task. These findings suggest that somatosensory information processing is regulated in a task- and environment-dependent manner. In this study, while we found that the SEP gating of the N33 component was more pronounced during the force task than the position task, more SEP gating of the P45 component was observed during the position task [14]. The N33 component of SEPs can reflect activation of area 1 [55,56]. Furthermore, in the non-human primate cortex, nearly all neurons in area 1 respond to cutaneous stimulation (90% or more), whereas areas 3a and 2 primarily respond to proprioceptive stimulation (for a review see [57]). Therefore, the difference in hierarchical processing between cutaneous and proprioceptive afferents in the primary somatosensory cortex may explain the difference between our present and previous findings. The physiological significance of SEP gating is considered to filter the sensory input that is not related to movement execution in order to improve the signal-to-noise ratio and to perform fine movements [23,54]. In the present study, it is reasonable to assume that, since changes in the length of muscle and joint angle are minimal during the force task, maintaining a constant force requires cutaneous more than proprioceptive information. Thus, the gating of the N33 component during the force task may reflect suppression of the sensory input that is not related to movement execution in order to maintain a constant muscle strength, and receive more pressure information to be applied to the force sensor. In other words, it is thought that SEP gating contributes to the effective use of neural resources [52,58,59]. Similarly, Staines et al. [60] revealed that amplitudes of SEPs from cutaneous afferents (sural nerve) and muscle afferents (tibial nerve) can be selectively modulated depending on the somatosensory information required for the task. These series of studies propose that amplitudes of SEP components are modulated by modality-specific manner depending on the somatosensory feedback needed for task performance. Alternatively, our finding may be ascribed to the centripetal mechanism. When changing the applied force, relatively more information regarding the joint angle and muscle length can be provided to the central nervous system during the position task, while relatively more information about the contact surface occurring at the skin can be provided to the central nervous system during the force task.

An interesting finding of this study was that the E2 amplitude of CMR was significantly greater during the force than position task. In previous studies, modulation of CMR amplitude with voluntary movements occurred primarily in the E2 component and depended on the context of the motor task [21,31,39,40]. Tuner et al. [21] demonstrated that the E2 amplitude of CMR recorded from the FDI muscle and the amplitude of SEPs (N20/P25) were smaller

during an isometric index finger abduction with self-paced tapping than the isometric contraction alone. They concluded that the reduction of the E2 amplitude resulted from gating of the digital nerve input. In the present study, gating of the SEP amplitude was larger during the force than position task, possibly indicating that the greater amplitude of the E2 component during the force task is not directly associated with the amount of skin afferent input. Meanwhile, the amplitude of the E2 component recorded from the FDI muscle was found to depend on the level of bEMG activity [40]; however, we found no significant difference in the bEMG activity between the position and force tasks. An alternative explanation for our finding could be a difference in cortical demands between the two tasks. Gibbs et al. [61] demonstrated that the E2 amplitude of CMR recorded from the tibialis anterior and soleus muscles was larger during voluntary activation of these muscles than quiet standing. This result implies that the E2 component can be larger during voluntary movements, which are controlled mainly by supraspinal pathway, as compared to during postural maintenance, which is mainly controlled at the spinal level. Thus, because cortical demands have been demonstrated to be greater in the force than position task [62,63], it is possible that the greater cortical demand during the force task resulted in the greater E2 amplitude of CMR. In addition, the task-dependent facilitatory and inhibitory modulations of the E2 component are supposed to contribute to the successful task performance by regulating muscles responses to cutaneous inputs [21,31,39,40,42]. For example, the E2 amplitude was enhanced during fatiguing contraction, indicating a compensation for decreased force producing capacity during the fatiguing contraction [42]. Therefore, the greater amplitude of the E2 component during the force task observed in this study may have contributed to the fast regulation of force via the cutaneomuscular reflex circuit. Additional studies are needed to clarify the more detailed relationship between the CMR and force control.

A limitation of this study is that EMG recording was limited to the FDI. Although we confirmed that activity level of the FDI muscle was similar between the force and postural tasks, we cannot completely deny a possibility that activities of the other auxiliary muscles, such as the finger extensor and palmar interosseous muscles, could have caused the difference in sensory gating between the two tasks, because sensory gating could increase with an increase in muscle activity [51].

Before closing, we would like to discuss potential clinical implications of our findings. Although isometric exercises are helpful for patients with central nervous disorders, exercise prescriptions are commonly provided without consideration of a difference in load type. In an animal study, cutaneous inputs are suppressed at the spinal level during a task that combines two load types [64]. On the other hand, in the present study that examined two load types separately, we found that gating of the N33 component was greater during the force than position task, which possibly indicates a relatively high use of task-related cutaneous information in the force task. Therefore, maintaining a constant force by pushing against a rigid restraint may be useful to improve the cutaneous information processing in patients with cutaneous sensory impairments. Furthermore, the neural circuits of CMR are known to be modulated by a short period of motor learning [46], a long-term badminton training [65], and an extensive piano training [66]. Thus, although more detailed studies are needed, repeated training may lead to improvements in force control.

In conclusion, we demonstrated that the reduction of the N33 amplitude of SEPs during the force task was significantly larger than that during the position task. Furthermore, the E2 amplitude of CMR was significantly greater during the force than position task. These findings suggest that cutaneous information processing differs with the load type during static muscle contraction.

## Supporting information

**S1 Dataset. Raw data.**
(XLSX)

## Acknowledgments

We would like to thank study participants who devoted their time and efforts.

## Author Contributions

**Conceptualization:** Hikari Kirimoto.

**Data curation:** Keisuke Yunoki.

**Formal analysis:** Keisuke Yunoki.

**Funding acquisition:** Tatsunori Watanabe, Takuya Matsumoto, Hikari Kirimoto.

**Investigation:** Keisuke Yunoki, Takuya Matsumoto, Takayuki Kuwabara, Takayuki Horinouchi, Kanami Ito, Haruki Ishida.

**Methodology:** Keisuke Yunoki, Hikari Kirimoto.

**Project administration:** Keisuke Yunoki, Hikari Kirimoto.

**Resources:** Keisuke Yunoki, Takuya Matsumoto, Takayuki Kuwabara, Takayuki Horinouchi, Kanami Ito, Haruki Ishida.

**Supervision:** Hikari Kirimoto.

**Validation:** Keisuke Yunoki.

**Visualization:** Keisuke Yunoki.

**Writing – original draft:** Keisuke Yunoki, Tatsunori Watanabe, Hikari Kirimoto.

**Writing – review & editing:** Keisuke Yunoki, Tatsunori Watanabe, Hikari Kirimoto.

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
