## [Decision Letter · Decision Letter 0]

1 Nov 2022

PONE-D-22-26508Cutaneous information processing differs with load type during isometric finger abductionPLOS ONE

Dear Dr. Kirimoto,

Thank you for submitting your manuscript to PLOS ONE. After careful consideration, we feel that it has merit but does not fully meet PLOS ONE’s publication criteria as it currently stands. Therefore, we invite you to submit a revised version of the manuscript that addresses the points raised during the review process. For acceptance, it is crucial that you discuss the physiological background of the N33 attenuation in more detail, as pointed out by the reviewer. In contrast, since novelty is not part of the publilcation criteria of PLOS ONE, it is recommended, but not essential to claim the difference from the results of Gibbs et al. (cf. point 3. of the review).

We look forward to receiving your revised manuscript.

Kind regards,

Peter Schwenkreis

Academic Editor

PLOS ONE

Journal Requirements:

2. Please note that PLOS ONE has specific guidelines on code sharing for submissions in which author-generated code underpins the findings in the manuscript. In these cases, all author-generated code must be made available without restrictions upon publication of the work. Please review our guidelines at https://journals.plos.org/plosone/s/materials-and-software-sharing#loc-sharing-code and ensure that your code is shared in a way that follows best practice and facilitates reproducibility and reuse. New software must comply with the Open Source Definition.

Reviewers' comments:

Reviewer's Responses to Questions

**Comments to the Author**

1. Is the manuscript technically sound, and do the data support the conclusions?

Reviewer #1: Partly

2. Has the statistical analysis been performed appropriately and rigorously? 

Reviewer #1: Yes

3. Have the authors made all data underlying the findings in their manuscript fully available?

Reviewer #1: Yes

4. Is the manuscript presented in an intelligible fashion and written in standard English?

Reviewer #1: No

5. Review Comments to the Author

Reviewer #1: This study demonstrates that the primary somatosensory cortex (S1) and a digital muscle respond differently to electrical stimulation of digital cutaneous nerves during different motor tasks. The authors used two isometric tasks in their experiments: a force task in which the index finger was pressed against a virtual wall with a constant force, and a position task in which the index finger was held stationary while lifting the same weight. The results showed that the N33 responses in the S1 were attenuated and cutaneous-muscular reflexes were enhanced in the force task rather than the position task. The authors argued that these results indicate that somatosensory information processing differs between these two isometric tasks. However, I believe that there is insufficient data and discussion to appeal such a claim.

1. The authors argued that cutaneous information processing differs based on differences in somatosensory evoked potentials in response to cutaneous nerve stimulation. However, the manuscript does not discuss this result in depth, and the physiological implications derived from this result are minimal. In particular, only N33 was attenuated in this study, but the authors did not explain what physiological significance lies behind this. It is also necessary to consider why N33 is attenuated while other components remain unchanged. Simply stating that cutaneous information processing differs depending on the isometric tasks is not novel compared to previous studies.

2. The authors strongly argued that the greater the attenuation of the SEP, the more sensory input is needed under those conditions. However, in my view, a decrease in S1 response means that less sensory information reaches the S1, i.e., that the S1 does not need somatosensory signals from the periphery. Therefore, the authors should provide more evidence or explanation why more sensory information is needed under conditions of greater SEP attenuation.

3. The result that cutaneous-muscular reflexes are enhanced has already been described in a paper (Gibbs et al 1995). Therefore, this study lacks novelty; it is necessary to claim the difference from the results of Gibbs et al.

4. The authors only explain the isometric tasks, and there is little background information to consider why differences in SEPs occur. Since the authors measured muscle activity, they should show muscle activity between the two tasks. Since in the position task co-contraction of agonistic muscle pairs occurs, it is inferred that whole muscle activity is larger in the position task than the force task. This would be consistent with a previous study that showed that greater muscle activity is associated with greater sensory gating (Sugawara et al., 2016). We would like to know if the differences in SEP modulation simply reflect differences in the magnitude of muscle activity.

5. Introduction is not easy to understand.

1. In the first paragraph, the authors described that it is difficult to evaluate Ia and Aβ separately (lines 63-65). However, in the third paragraph, they described that many experiments have been done to stimulate cutaneous nerves (lines 79-82). These sentences are not consistent. The text should be rearranged.

2. In the first sentence of the second paragraph, the authors described that modality specific information processing is being investigated. But, in fact, only muscular information processing is mainly mentioned in the paragraph.

3. Before the last sentence of the second paragraph, it is necessary to state the reason why cutaneous information processing is examined in the position task and the force task.

Minor points

Lines 216-218; The authors should be more specific about the measurement of SEP.

Lines 295-301; Doesn't the greater attenuation of SEP to peripheral stimuli mean that the central nervous system does not need that information?

Lines 337-347; Since the visual feedback is the same in the two tasks, there is no need to discuss the influence of visual feedback on somatosensory processing. This paragraph should be deleted.

Does the noise affect the results because the SEP signal is noisy in Figure 3. Does applying a 200 Hz low pass filter not change the results?

6. PLOS authors have the option to publish the peer review history of their article (what does this mean?). If published, this will include your full peer review and any attached files.

Reviewer #1: No

---

## [Author Response · Author response to Decision Letter 0]

15 Nov 2022

Dear Editor and Reviewer 

Thank you very much for reviewing our manuscript and offering valuable advice.

We have addressed all the comments with point-by-point responses and revised the manuscript accordingly.

1. The authors argued that cutaneous information processing differs based on differences in somatosensory evoked potentials in response to cutaneous nerve stimulation. However, the manuscript does not discuss this result in depth, and the physiological implications derived from this result are minimal. In particular, only N33 was attenuated in this study, but the authors did not explain what physiological significance lies behind this. It is also necessary to consider why N33 is attenuated while other components remain unchanged. Simply stating that cutaneous information processing differs depending on the isometric tasks is not novel compared to previous studies.

Response: Thank you for raising an important point. We have added an explanation of the physiological implications of the finding that N33 amplitude was attenuated to the Discussion (Page 12-13, Line 305-312).

2. The authors strongly argued that the greater the attenuation of the SEP, the more sensory input is needed under those conditions. However, in my view, a decrease in S1 response means that less sensory information reaches the S1, i.e., that the S1 does not need somatosensory signals from the periphery. Therefore, the authors should provide more evidence or explanation why more sensory information is needed under conditions of greater SEP attenuation.

Response: We appreciate your suggestion. A decrease in SEP amplitude during the motor task compared to rest, that is, SEP gating, does not imply decreased S1 excitability. The functional significance of SEP gating is thought to be that it suppresses nerve excitability caused by electrical stimulation, which is necessary for recording SEPs but constitutes a source of noise during task execution. Please refer to Page 4, Lines 86 to 89 for this content.

3. The result that cutaneous-muscular reflexes are enhanced has already been described in a paper (Gibbs et al 1995). Therefore, this study lacks novelty; it is necessary to claim the difference from the results of Gibbs et al.

Response: We agree that the result that the E2 amplitude of CMR was larger in muscles performing voluntary contraction than in posturally active muscles has been reported by Gibbs et al. However, we suggest that our study has novelty in the following two points. 

1. Gibbs et al compared the amplitude of CMR recorded from lower and trunk muscles following electrical stimulation of the digital nerves of the second toe between the voluntary task (subjects lay supine or prone and voluntarily contracted lower limb and trunk muscles) and postural task (subjects maintained an upright posture). However, in the present study, we compared the amplitude of CMR between two tasks that differed only in pure load type. 

2. We simultaneously measured SEPs and found that cortical responses to cutaneous information differed between the two tasks.

4. The authors only explain the isometric tasks, and there is little background information to consider why differences in SEPs occur. Since the authors measured muscle activity, they should show muscle activity between the two tasks. Since in the position task co-contraction of agonistic muscle pairs occurs, it is inferred that whole muscle activity is larger in the position task than the force task. This would be consistent with a previous study that showed that greater muscle activity is associated with greater sensory gating (Sugawara et al., 2016). We would like to know if the differences in SEP modulation simply reflect differences in the magnitude of muscle activity.

Response: Thank you for pointing out an important aspect. We have noted that subjects performed the position and force tasks for approximately 90 s (divided into three blocks of 30 s) each before recording the SEPs and CMR, and confirmed that EMG activity of the right FDI muscle was equal between both tasks in the sub-section of “Protocol” (Page 7, Line 172-174). Furthermore, we monitored the FDI EMG activity during the recordings of SEPs and CMR to ensure that the EMG activity was equal between the two tasks. This explanation has been added to the manuscript (Page 8, Line 186-187).

5. Introduction is not easy to understand.

1. In the first paragraph, the authors described that it is difficult to evaluate Ia and Aβ separately (lines 63-65). However, in the third paragraph, they described that many experiments have been done to stimulate cutaneous nerves (lines 79-82). These sentences are not consistent. The text should be rearranged.

Response: Thank you for pointing out that our statement could be understood in this way. In daily life, both Aβ and Ia fibers are activated by movements and these two kinds of information are important for precise movement execution. Therefore, it is difficult to set up an assignment that isolates their respective contributions. In the present study, we addressed this issue by using isometric contraction tasks with different load types under experimental environments. We have rearranged these sentences (Page 3, Line 63-65).

2. In the first sentence of the second paragraph, the authors described that modality specific information processing is being investigated. But, in fact, only muscular information processing is mainly mentioned in the paragraph.

Response: Thank you for pointing this out. We evaluate not only muscle information but also reflex responses of the spinal cord or central nervous system to nerve stimuli (especially group Ia fibers) by EMG. We highlighted that only proprioceptive information processing has been found to differ between the position and force tasks (Page 3, Line 66-67).

3. Before the last sentence of the second paragraph, it is necessary to state the reason why cutaneous information processing is examined in the position task and the force task.

Response: As pointed out, we have described the importance of examining differences in cutaneous information processing between the position and force tasks (Page 4, Line 78-82). 

Minor points

Lines 216-218; The authors should be more specific about the measurement of SEP.

Response: Thank you for your helpful advice. We have provided a more detailed description of the analysis of SEP data (Page 9, Line 224-225 and 227-228). Further descriptions of SEP measurement can be found in the sub-section “Recordings of SEPs”. 

Lines 295-301; Doesn't the greater attenuation of SEP to peripheral stimuli mean that the central nervous system does not need that information?

Response: This is correct. We have inserted material about the functional role of SEP gating into the Discussion (Page 13, Line 312-321)

Lines 337-347; Since the visual feedback is the same in the two tasks, there is no need to discuss the influence of visual feedback on somatosensory processing. This paragraph should be deleted.

Response: As pointed out, we have deleted that paragraph.

Does the noise affect the results because the SEP signal is noisy in Figure 3. Does applying a 200 Hz low pass filter not change the results?

Response: We appreciate your suggestion. While we agree that restricting the low-pass filter would reduce the noise, we prefer to keep the low-pass filter in its present configuration because restriction would also affect the amplitude of the short latency components of SEP (Desmedt et al., 1974; Shaw, 1992). As you pointed out, the SEP raw waveform in Fig. 3 may look a little noisy, but this presents no problems for measuring the important amplitude values.

---

## [Decision Letter · Decision Letter 1]

5 Dec 2022

PONE-D-22-26508R1Cutaneous information processing differs with load type during isometric finger abductionPLOS ONE

Dear Dr. Kirimoto,

Thank you for submitting your manuscript to PLOS ONE. After careful consideration, we feel that it has merit but does not fully meet PLOS ONE’s publication criteria as it currently stands. Therefore, we invite you to submit a revised version of the manuscript that addresses the points raised during the review process.

We look forward to receiving your revised manuscript.

Kind regards,

Peter Schwenkreis

Academic Editor

PLOS ONE

Journal Requirements:

Reviewers' comments:

Reviewer's Responses to Questions

**Comments to the Author**

1. If the authors have adequately addressed your comments raised in a previous round of review and you feel that this manuscript is now acceptable for publication, you may indicate that here to bypass the “Comments to the Author” section, enter your conflict of interest statement in the “Confidential to Editor” section, and submit your "Accept" recommendation.

Reviewer #1: All comments have been addressed

2. Is the manuscript technically sound, and do the data support the conclusions?

Reviewer #1: Yes

3. Has the statistical analysis been performed appropriately and rigorously? 

Reviewer #1: Yes

4. Have the authors made all data underlying the findings in their manuscript fully available?

Reviewer #1: Yes

5. Is the manuscript presented in an intelligible fashion and written in standard English?

Reviewer #1: Yes

6. Review Comments to the Author

Reviewer #1: The revised version has been improved in terms of quality of introduction compared to the original submission. However, some points remain to be revised.

Related to comment 2:

Although the function of sensory gating is said to reduce irrelevant information, the authors’ experiment did not show whether the sensory information was needed in such a situation. Without careful explanation, it is shortsighted to argue that a situation indicating the greater SEP gating requires more sensory information. The related sentences should be removed (lines 92-95, 286-287).

Related to comment 3:

The authors should add a discussion which argues a difference between a previous paper about the enhancement of cutaneous-muscular reflexes (Gibbs et al 1995), as the authors responded as the first point.

Related to comment 4:

The author stated that the position and force tasks require a similar net muscle torque (Lines 70-71). However, they only recorded the activity of a single muscle (FDI). We do not know whether the total activity of the whole hand muscles is equivalent between two tasks. A previous study showed that greater muscle activity is associated with greater sensory gating (Sugawara et al., 2016). The difference in sensory gating between the two tasks may simply reflect differences in the magnitude of muscle activity. As it stands, this paper may send the wrong message. Therefore, the authors should add a discussion of the possibility that differences in muscle activity that are not recorded cause differences in sensory gating between two tasks.

7. PLOS authors have the option to publish the peer review history of their article (what does this mean?). If published, this will include your full peer review and any attached files.

Reviewer #1: No

---

## [Author Response · Author response to Decision Letter 1]

6 Dec 2022

Dear Editor and Reviewer 

Thank you very much for reviewing our manuscript and offering valuable advice.

We have addressed all the comments with point-by-point responses and revised the manuscript accordingly.

1. Although the function of sensory gating is said to reduce irrelevant information, the authors’ experiment did not show whether the sensory information was needed in such a situation. Without careful explanation, it is shortsighted to argue that a situation indicating the greater SEP gating requires more sensory information. The related sentences should be removed (lines 92-95, 286-287).

Response: As pointed out, we have not demonstrated that the afferent information from the finger nerves is important or that the electrical stimulation to elicit SEP is noise. Thus, we have deleted the sentences you pointed out and revised several sentences in the Discussion (Page 13, Line 308-316; Page 15, Line 366-367).

2. The authors should add a discussion which argues a difference between a previous paper about the enhancement of cutaneous-muscular reflexes (Gibbs et al 1995), as the authors responded as the first point.

Response: As suggested, we have described in detail the results of Gibbs et al. in order to clarify the difference between the present and previous studies (Page 14, Line 340-344).

3. The author stated that the position and force tasks require a similar net muscle torque (Lines 70-71). However, they only recorded the activity of a single muscle (FDI). We do not know whether the total activity of the whole hand muscles is equivalent between two tasks. A previous study showed that greater muscle activity is associated with greater sensory gating (Sugawara et al., 2016). The difference in sensory gating between the two tasks may simply reflect differences in the magnitude of muscle activity. As it stands, this paper may send the wrong message. Therefore, the authors should add a discussion of the possibility that differences in muscle activity that are not recorded cause differences in sensory gating between two tasks.

Response: Thank you for pointing out an important aspect. We have discussed about this point in the Discussion (Page 14-15, Line 355-360).

---

## [Editor Report · Decision Letter 2]

8 Dec 2022

Cutaneous information processing differs with load type during isometric finger abduction

PONE-D-22-26508R2

Dear Dr. Kirimoto,

We’re pleased to inform you that your manuscript has been judged scientifically suitable for publication and will be formally accepted for publication once it meets all outstanding technical requirements.

Kind regards,

Peter Schwenkreis

Academic Editor

PLOS ONE
---

## [Editor Report · Acceptance letter]

14 Dec 2022

PONE-D-22-26508R2 

Cutaneous information processing differs with load type during isometric finger abduction 

Dear Dr. Kirimoto:

I'm pleased to inform you that your manuscript has been deemed suitable for publication in PLOS ONE. Congratulations! Your manuscript is now with our production department. 

Kind regards, 

on behalf of

Dr. Peter Schwenkreis 

Academic Editor

PLOS ONE